# Concurrent Chronic Exertional Compartment Syndrome and Popliteal Artery Entrapment Syndrome

**DOI:** 10.3390/diagnostics14161825

**Published:** 2024-08-21

**Authors:** Tiffany R. Bellomo, Connie Hsu, Pavan Bolla, Abhisekh Mohapatra, Dana Helice Kotler

**Affiliations:** 1Division of Vascular and Endovascular Surgery, Massachusetts General Hospital, Boston, MA 02114, USA; pbolla519@gmail.com (P.B.); amohapatra@mgh.harvard.edu (A.M.); 2Division of Orthopedic Surgery, Massachusetts General Hospital, Boston, MA 02114, USA; cmhsu@mgh.harvard.edu (C.H.); dkotler@mgh.harvard.edu (D.H.K.)

**Keywords:** chronic exertional compartment syndrome, popliteal artery entrapment syndrome, athletic leg pain

## Abstract

Exertional leg pain occurs with notable frequency among athletes and poses diagnostic challenges to clinicians due to overlapping symptomatology. In this case report, we delineate the clinical presentation of a young collegiate soccer player who endured two years of progressive bilateral exertional calf pain and ankle weakness during athletic activity. The initial assessment yielded a diagnosis of chronic exertional compartment syndrome (CECS), predicated on the results of compartment testing. However, her clinical presentation was suspicious for concurrent type VI popliteal artery entrapment syndrome (PAES), prompting further radiographic testing of magnetic resonance angiography (MRA). MRA revealed severe arterial spasm with plantarflexion bilaterally, corroborating the additional diagnosis of PEAS. Given the worsening symptoms, the patient underwent open popliteal entrapment release of the right leg. Although CECS and PAES are both known phenomena that are observed in collegiate athletes, their co-occurrence is uncommon owing to their different pathophysiological underpinnings. This case underscores the importance for clinicians to be aware that the successful diagnosis of one condition does not exclude the possibility of a secondary, unrelated pathology. This case also highlights the importance of dynamic imaging modalities, including point-of-care ultrasound, dynamic MRA, and dynamic angiogram.

## 1. Introduction

Exertional leg pain, prevalent among athletes [1], is characterized by pain distal to the knee and proximal to the talocrural joint, exacerbated by physical activity [2]. The multifaceted etiology of exertional leg pain encompasses various pathologies. A comprehensive study of athletes experiencing exertional leg pain revealed diverse diagnoses: chronic exertional compartment syndrome (CECS) in 33%, stress fractures in 25%, medial tibial stress syndrome (MTSS) in 13%, and entrapment syndromes such as popliteal artery entrapment syndrome (PAES) in 10% [3]. With regards to incidence, CECS has been described as occurring up to 0.49 cases per 1000 person-years [4]. PAES is even rarer at an incidence of 0.16 to 3.5 cases per 100,000 person-years [5]. Diagnostic approaches to these diseases include physical exam maneuvers and various imaging modalities. In this specific case, dynamic imaging modalities were the most elucidating in the assessment of external vascular compression, vessel wall disease, and musculoskeletal overuse syndromes. While static computed tomography angiography (CTA) was the prior screening and diagnostic tool of choice to assess for compression, ultrasound imaging, dynamic angiography, and dynamic MRA have proven their utility. An accurate diagnosis of exertional leg pain poses challenges due to overlapping clinical presentations, yet it remains imperative given the spectrum of treatment modalities that range from medical management to operative intervention. Concurrent diagnoses do not often occur but can certainly make management challenging. While concurrent diagnoses are infrequent, their occurrence can present formidable management obstacles. In this case report, we present the clinical scenario of a young collegiate soccer player experiencing progressive bilateral exertional calf pain over a two-year period, ultimately diagnosed with both CECS and PAES bilaterally, necessitating surgical intervention. 

## 2. Case Presentation 

### 2.1. Initial Presentation

The patient described in this report is a 21-year-old female with no past medical history who initially presented with 2 years of bilateral calf pain. Her physical activity was notable for intense competitive soccer training at the collegiate level. At the onset of symptoms, she noticed mild bilateral calf pain a few hours after playing soccer (Figure 1).

At that time, her athletic trainer noted her calves were very tight and recommended manual therapy and friction massage therapy, which temporarily abated her symptoms. However, the pain subsequently progressed to a 10 out of 10 burning sensation in the posterior bilateral calves with less than 10 min of running. In addition, she developed associated bilateral weakness with ankle dorsiflexion, plantarflexion, and eversion. She denied paresthesias with exertion and at rest. She denied other joint pain, chronic fatigue, and other systemic symptoms of infection, including fevers and chills. She had never had surgery before. She had never smoked or taken any illicit drugs. Her family history was unremarkable. 

On physical examination, her spine and lower extremity alignment were normal. Her bilateral lower extremities were grossly normal. She had a full range of motion and sensation was intact to light touch. She had 5/5 strength throughout bilaterally, including hip flexion, knee flexion, knee extension, ankle dorsiflexion, ankle plantarflexion, eversion, and extensor hallucis longus strength. Active and passive ankle movements elicited pain during examination. She did not have tenderness to palpation of the bilateral paraspinal, gluteal, hamstring, or gastrocnemius/soleus muscles. Tinel’s test performed around the fibular head was negative bilaterally. She reported discomfort with heel walking and toe walking. Single leg squat was normal bilaterally. She had no pain with hopping on one foot or the other. In terms of vascular findings, she had 2+ palpable dorsalis pedis and posterior tibial pulses. Ishikawa’s sign was negative, as dorsalis pedis and posterior tibial pulses remained palpable with dorsiflexion and plantarflexion of the ankle. She had no evidence of varicose veins. There was no edema or lower extremity wounds.

### 2.2. Diagnostic Testing 

The differential for bilateral leg pain on exertion is very broad, including external vascular compression (PAES, CECS, adductor canal compression syndrome, and cystic adventitial disease), vessel wall disease (peripheral arterial disease [PAD] and arterial endofibrosis), musculoskeletal overuse (medial tibial stress syndrome, tibial stress fracture, and soleal sling syndrome), and nervous system origin (nerve entrapments, lumbar disc herniation, and complex regional pain syndrome). Given the symptoms of burning in bilateral posterior calves with exercise that stopped after rest and the age of this young female soccer player, both categories of external vascular compression and musculoskeletal overuse were considered. 

X-rays were performed initially to assess for musculoskeletal overuse sequelae, including stress fractures, which did not show any evidence of overt or hairline fracture bilaterally. Consequently, orthopedic specialists assessed for external vascular compression by measuring compartment pressures. This testing revealed elevated pressures reaching up to 60 mmHg subsequent to treadmill running at a local orthopedic facility, thereby corroborating the diagnosis of CECS (Table 1). 

Concurrently, the manifestation of discomfort during toe walking and plantar flexion raised suspicions of PAES. A point-of-care ultrasound conducted during a sports medicine clinic revealed a patent popliteal artery in the neutral position that was compressed after plantar flexion, which was more pronounced in the right leg than the left leg. To further elucidate these findings, an ankle brachial index (ABI) obtained was normal bilaterally at rest, and therefore, intrinsic vessel wall disease was less likely. Dynamic magnetic resonance angiography (MRA) of the bilateral lower extremities was also obtained (Figure 2). 

There was evidence of excess muscle in the intercondylar notch bilaterally, greater hypertrophy of the medial head of the gastrocnemius on the right more than the left, and compression of the popliteal arteries and veins during plantarflexion. Although these were positive findings, the extent of arterial compression was not fully appreciated. Therefore, she underwent a diagnostic angiogram with vascular surgery that demonstrated dynamic right popliteal artery compression during plantarflexion bilaterally, confirming the diagnosis of PEAS (Figure 3).

### 2.3. Definitive Treatment 

Following the patient’s initial consultation, she presented to the emergency department with new-onset right medial knee, right-sided calf pain, and visible bruising and edema without any inciting trauma. Evaluation for deep venous thrombosis was negative. Given the progressive nature of symptoms attributed to both chronic exertional compartment syndrome (CECS) and popliteal artery entrapment syndrome (PAES), vascular surgery recommended operative popliteal entrapment release of the right leg. 

The operation commenced with a posterior lazy S incision in the right popliteal fossa. The dissection was carried down to the popliteal artery with visualization of the gastrocnemius muscle. In terms of anatomical findings, the gastrocnemius muscle appeared to be severely hypertrophied and there were no bands of aberrant gastrocnemius muscle attached to the tibia. The popliteal artery was in the normal anatomic position, where the artery coursed adjacent to and lateral to the medial head of the gastrocnemius muscle. Therefore, PAES type VI was diagnosed, and the medial head of the gastrocnemius muscle was resected. After this release, provocative maneuvers no longer compressed the proximal popliteal artery intraoperatively. The patient was made immediately weight bearing post-operative day one and subsequently discharged with physical therapy clearance on post-operative day two. At a one-month follow-up appointment, the patient reported improvement of pain in her right calf with provocative maneuvers and is currently undergoing rehabilitation.

## 3. Discussion

This case presents a unique occurrence wherein a young female athlete exhibited bilateral exertional leg pain and was concurrently diagnosed with CECS and type VI PAES (Table 1). In this discussion, we will provide a broad overview of other diagnoses that present with bilateral leg pain, including external vascular compression (PAES, CECS, adductor canal compression syndrome, and cystic adventitial disease), vessel wall disease (PAD and arterial endofibrosis), musculoskeletal overuse (medial tibial stress syndrome, tibial stress fracture, and soleal sling syndrome), and nervous system origin (nerve entrapments, lumbar disc herniation, and complex regional pain syndrome). The differential for this specific case was narrowed by dynamic imaging modalities, including point-of-care ultrasound, dynamic MRA, and dynamic angiogram. This case underscores the importance for clinicians to be aware that the successful diagnosis of one condition via diagnostics does not exclude the possibility of a secondary pathology.

### 3.1. External Vascular Compression

Among the diagnoses for bilateral leg pain within the category of external vascular compression, CECS is common in young athletes and is characterized by elevated pressures within the muscular compartments following exercise [6]. Symptoms are often bilateral and consist of pain described as a cramp or fullness. Although peripheral neuropathies are uncommon, there may be associated numbness, tingling, and weakness in the peroneal or sural nerve distribution [7]. The pathophysiology is thought to be reversible, exercise-induced compression within the fascial compartment, causing thickening and scarring of the gastrocnemius and soleal fascia. These changes inhibit the capacity of the muscles for expansion to accommodate increased arterial flow with exercise, resulting in diminished tissue perfusion and subsequent ischemic pain across any of the four compartments of the lower leg [3]. Needle manometry, or compartment testing, serves as the gold standard diagnostic procedure for CECS. Compartment testing involves the measurement of intercompartmental pressures in the affected extremity at rest, during exercise, and after exercise. Pressures in the symptomatic compartment are usually greater than 25 mm [7]. The initial treatment typically involves conservative measures such as massage, rest, and gait training. Gait retraining involves changing the patient’s pattern to a forefoot strike, which may improve anterior compartment symptoms in runners [8]. For patients with refractory symptoms, fasciotomy is the standard of care [9].

Within the category of external vascular compression, PAES represents a less common yet potentially limb-threatening vascular and musculoskeletal condition seen in 0.17–3.5% of the general US population [10,11,12]. PAES denotes a specific anatomic variation wherein the popliteal artery is compressed by a surrounding myofascial structure with exercise [6]. Depending on the course of the popliteal artery and compressing structure, PAES is classified into six different variations, as proposed by Love and Whelan and modified by Rich et al. [11]. Type I PAES specifies that the popliteal artery is positioned medial and deep to the gastrocnemius muscle or tendon. The aberrant medial arterial course allows for compression of the popliteal artery by the normally placed medial head of the gastrocnemius muscle [13]. Type II denotes that the popliteal artery is compressed by the medial head of the gastrocnemius muscle, emanating from a lateral position. In this type, the abnormal medial head of the gastrocnemius inserts laterally on the distal femur, medially displacing the popliteal artery. Type III denotes that the popliteal artery remains in its normal position, but an aberrant accessory slip or tendon from the medial head of the gastrocnemius muscle wraps around and entraps the popliteal artery [9]. Type IV denotes that the popliteal artery is compressed by the popliteus muscle. Type V denotes that both the popliteal artery and vein are compressed, involving any form of the first four types, but includes compression of the popliteal vein, as well. Type VI is also known as functional PAES, where the popliteal artery is compressed due to muscle hypertrophy. While both type IV PAES and CECS result in ischemic pain during exercise, PAES involves the compression of the popliteal artery due to a hypertrophied gastrocnemius muscle, which is a distinct but related pathophysiological mechanism compared to CECS [7]. 

While many individuals born with the PAES congenital anomaly remain asymptomatic, endurance activities, notably running, can induce changes in the gastrocnemius muscle, leading to muscle hypertrophy and, subsequently, popliteal artery impingement by the medial head of the gastrocnemius [14]. Clinically, this presents as painful claudication-type symptoms in otherwise healthy individuals. The gold standard diagnostic study is both MRA and angiography, demonstrating popliteal artery impingement [14]. Management of PAES often requires surgical exploration and release of the popliteal artery through sectioning of fibrous bands [15]. Left untreated, PAES may result in complications such as popliteal artery stenosis, thrombosis, or limb-threatening distal arterial thromboembolism. While CECS and PAES are known phenomena observed in collegiate athletes, their simultaneous occurrence is atypical due to the distinct mechanisms of action delineated above. Specifically, in this patient, it was suspected that these diagnoses were related. One hypothesis is that the patient’s venous and arterial compression led to the development of exertional venous hypertension and associated muscle edema that presented as exertional leg pain. Therefore, it is possible that transient ischemia and reperfusion injury caused by PAES contributed to the additional edema and exacerbated a pre-existing CECS. 

Other categories of external vascular compression include adductor canal compression syndrome, involving the entrapment of the superficial femoral artery by fibrous bands from the adductor magnus muscle or a hypertrophied vastus medialis or adductor magnus muscle [16,17]. This condition typically presents as intermittent claudication in young, physically active individuals, such as runners and skiers. Symptoms like paleness, coldness, or weak peripheral pulses are rare in the early stages, making patient history crucial for diagnosis in addition to magnetic resonance imaging (MRI). The condition can lead to limb-threatening ischemia if left untreated [18]. Another condition includes cystic adventitial disease, which is a nonatherosclerotic condition characterized by cyst formation between the adventitia and tunica media of arteries, most commonly the popliteal artery [19]. These cysts, which can be unilocular or multilocular and may contain mucin, are often present in males in their fourth and fifth decades of life. The disease manifests similarly to PAES but includes Ishikawa’s sign or loss of pedal pulses with knee flexion. Duplex ultrasonography is typically used for diagnosis. In the presented case, the MRI performed did not show hypertrophy of other muscles suggestive of adductor canal compression syndrome, and Ishikawa’s sign was negative, making cystic adventitial disease less likely. 

### 3.2. Vessel Wall Disease

Peripheral arterial disease (PAD) is a vascular condition marked by atherosclerosis, leading to vascular obstruction [20]. Common risk factors include smoking, dyslipidemia, poor diabetic control, and hypertension [21]. Patients with PAD often experience intermittent claudication and may present with a non-palpable distal pulse. An ankle brachial index of less than 0.9 is diagnostic of PAD. Treatment often depends on the severity of the disease, the presence of rest pain, and signs of tissue loss [22]. In the specific case of the young female nonsmoker presented in this report, she did not have any significant risk factors for PAD, and her resting ABI/PVRs were normal.

Another disease affecting the arterial wall includes arterial endofibrosis, which primarily presents in the external iliac artery [23]. This disease, common in cyclists, is characterized by leg weakness and thigh pain on exertion due to loose connective tissue buildup within the arterial walls, unlike the atherosclerotic plaques in PAD. Diagnosis is challenging, with digital subtraction angiography being the most reliable method [24]. CT angiography can also reveal stenosis, but minimal endofibrosis may not be detected on imaging [25]. Although endofibrosis was possible in this case, the patient did not have prominent thigh pain, and diagnostic angiography did not show stenosis within the external iliac arteries. 

### 3.3. Musculoskeletal Overuse 

MTSS is characterized by pain along the posterior medial border of the tibia, typically due to overuse [26]. Common in athletes, the pain initially decreases with activity but worsens with continued endurance training. Physical exam findings include pain on resisted flexion, and radiological tests are used to rule out stress fractures [27]. Stress fractures in the tibia can also occur from sudden increases in activity levels, leading to a gradual onset of pain that worsens over time and can occur at rest. Key signs include focal tenderness and swelling around the fracture site and a positive hop test. Initial X-rays may not detect the fracture, but MRI can also identify fractures. The young female athlete described here did not have pain on palpation over the tibia, and her pain did not improve after initial activity, making this diagnosis less likely. This patient was also assessed for stress fractures, with both X-ray and MRI. 

Soleal sling syndrome involves compression of the tibial nerve as it passes under the origin of the soleus, which becomes hypertrophied with repeated use [28]. Symptoms of this condition include plantar pain, numbness, and calf tightness without intermittent claudication. Physical examination reveals pain along the tibial nerve in the posterior knee, named the Tinel sign, and isolated flexor hallucis longus weakness [28]. On physical examination of the patient in this study, she had a negative Tinel’s test and no weakness in either lower extremity bilaterally. 

### 3.4. Nervous System Etiology 

Lumbar disc herniation is another source of leg pain, originating from the buttocks to the back of the leg [29]. The sharp, shooting pain is often accompanied by bilateral paresthesias. Symptoms worsen when seated due to nerve compression. A straight leg test can reproduce the symptoms, and MRI is the gold standard for diagnosis [30]. This patient had a negative straight leg test on examination, and MRI showed no evidence of disc herniation or degenerative disease. 

Nerve entrapments of the lower extremity may also mimic pain that is similar to claudication. However, this presentation also includes paresthesia in the nerve distribution, worsening with exertion. An examination may show reproducible pain when tapping or palpating the sites of compression and muscle weakness specific to the entrapped nerve. Diagnostic nerve blocks and electromyography testing can aid in diagnosis if there is a high suspicion [31]. The young patient, in this case, had a low diagnostic suspicion, given the absence of paresthesias. 

Complex regional pain syndrome involves severe, prolonged pain, often following an inciting injury [32]. Symptoms include intense pain, swelling, and changes in skin color and temperature. The diagnosis is clinical, based on the patient history and symptoms [33]. This diagnosis of exclusion was considered in this study but was quickly ruled out based on the dynamic imaging modalities used. 

### 3.5. Modalities of Imaging 

Advancements in diagnostic imaging have allowed for static and dynamic modalities that can narrow the differential diagnosis for bilateral leg pain. In this specific case, dynamic imaging modalities were the most elucidating in the assessment of external vascular compression, vessel wall disease, and musculoskeletal overuse syndromes. While static computed tomography angiography (CTA) was the prior screening and diagnostic tool of choice to assess for compression, ultrasound imaging, dynamic angiography, and dynamic MRA have proven their utility. 

The initial diagnostic tool to assess for PAES in this case was an in-office dynamic point-of-care ultrasound with color Doppler. Interestingly, the dynamic duplex ultrasound at select institutions has become a screening tool for all athletes with leg pain during exercise [34]. This screening tool is supported by a 10-year retrospective study of PAES, showing that 50% of patients with PAES had a positive duplex ultrasound, but 100% of patients with PAES had a positive dynamic duplex ultrasound for vascular compression [35]. More contemporary studies estimate that the dynamic duplex ultrasound had a 76% sensitivity for PAES. Another ultrasound-based study that provides substantial information includes dynamic exercise ABIs. A study by Morgan et al. noted that exercise ABIs were the most reliable vascular laboratory test to distinguish PAES from CECS [36]. However, both diagnoses can exist simultaneously, and compartment pressures should still be included in the diagnostic testing. 

Dynamic MRA can more clearly and definitively demonstrate not only arterial compression but also the anatomic muscular structures responsible for compression [34]. MRA, in general, can reveal pathological issues such as the abnormal positioning of the medial head of the gastrocnemius, the medial displacement and blockage of the popliteal artery, and fat tissue occupying the usual space of the medial head of the gastrocnemius [37]. Specifically, dynamic MRA can distinguish between anatomical and functional PAES and provide clear muscle and boundary details [34]. MRA does have disadvantages, including contraindications of metallic foreign bodies and the time needed for patients to hold active extension or active plantarflexion for the duration of the imaging time. 

Although MRA did demonstrate arterial compression in the case presented, the dynamic angiogram clearly showed severe compression of the popliteal arteries and veins during plantarflexion. In some cases, arterial compression is only visible during a dynamic angiogram [38]. This was quantified in a study by Ghaffarian et al., where four out of five protocolized MRA studies were falsely negative in patients who all had type III PAES [37]. Both US and MRA protocolized studies require experienced radiology technicians who can adequately instruct patients on provocative maneuvers, which may not be available at some institutions. The benefit of dynamic angiography is that patients only need to perform provocative maneuvers for 2 to 3 s under direct guidance by their vascular surgeon, and it was, therefore, the modality of choice for most physicians in a cohort study spanning over 11 years [34,35]. Radiation exposure is similar to a static CTA, exposing the patient on average to less than 4 miligrays [39]. 

### 3.6. Prognosis after Treatment 

The main treatment strategy for diseases within the external vascular compression category is to remove the external compressor. This can be done through either physical therapy or invasive surgical intervention. For CECS, the initial treatment of gait retraining involves changing the patient’s pattern to a forefoot strike [8,40]. Other conservative treatments involve rest and avoidance of the activity [41]. Given avoidance is not a favorable option for athletes and CECS can cause refractory symptoms, most patients undergo a fasciotomy to remove the external compression [9]. In a systematic review article outlining the outcomes of operative management, data from 24 studies, including 1596 patients, were extracted, and the outcomes were controversial [42]. The overall complications consisted of neuritis, infection, and hematoma at a rate of 13%. The rate of symptom improvement after compartment release for CECS was 66%, and 84% of patients were satisfied with their surgical outcomes in follow-up. Explanations for unsuccessful cases included no consensus on the definition of “success”, and a subset of patients with deep posterior compartment syndrome may derive less benefit from surgical release [4]. 

Conversely, the initial management of symptomatic PAES is surgical exploration and release of the popliteal artery through the sectioning of fibrous bands [15,43]. Left untreated, PAES may result in complications such as popliteal artery stenosis, thrombosis, or limb-threatening distal arterial thromboembolism. For PAES, surgical outcomes are generally positive: one recent retrospective study reported no complications and a success rate of 100% [44]. There is a higher rate of symptom improvement in PAES symptoms compared to CECS at 97% [45,46]. Complications such as post-operative hematoma, poor healing, and infections occurred in up to 16% of patients [45]. Long-term follow-up studies indicate that the primary patency rate is 98%. A significant portion of patients are able to resume sports, with about 33% returning fully, 50% partially, and 17% not resuming sports at the time of follow-up [17]. 

Vessel wall diseases are normally treated with aggressive conservative strategies. PAD is treated with rigorous walking therapy and risk factor modification [20]. If the disease becomes severe and the patient develops rest pain or an ulcer, an endovascular-first approach is generally used [47]. The prognosis of PAD, overall, varies widely depending on compliance, comorbidities, and disease severity. Severe PAD requiring amputation confers an increased risk of mortality, with around 50% of patients dying within one year of their amputation [48]. While arterial endofibrosis is also a vessel wall disease normally treated conservatively with position modification, this disease occurs in healthy cyclists [49]. The pathology is related to repetitive use, and most patients have a good prognosis [50]. 

The most common musculoskeletal overuse disease is MTSS, where a daily regimen of calf stretching and eccentric calf exercises is highly recommended and efficacious [51]. Other recommended treatments include shock-absorbing soles of shoes, insoles to prevent excessive foot pronation, and proprioceptive balance [52]. For patients with refractory disease, surgical treatment performed by removing a strip of periosteum produced an “excellent” result in only 35% of patients [53]. 

The treatment of leg pain originating from nervous system dysfunction varies widely depending on the pathology. While there is limited high-level evidence of a variety of conservative treatments, expert consensus agrees that a limited course of structured exercise is warranted for patients with lumbar disc herniation with radiculopathy [30]. There is a moderate level of evidence that supports discectomy for lumbar disc herniation with radiculopathy in patients whose straight leg test is positive [30]. Pain, as measured by the visual analog scale 6 months post-operatively, improved in over 70% of patients [54,55]. Treatments for nerve entrapments are mainly conservative: discontinuation of aggravating activities; use of local physical modalities such as ice, heat, or transcutaneous electrical nerve stimulation; desensitization techniques; correction of training errors that may be contributing to the nerve entrapment; braces; and nonsteroidal anti-inflammatory drugs may all be used [31]. The surgical approach for patients with refractory symptoms depends on the type of nerve entrapment. For peroneal nerve entrapment requiring surgical decompression, athletes usually return to normal activities of daily living within 3 weeks post-surgery [56]. Complex regional pain syndrome is another nervous system disorder that is primarily treated by targeting both exercise and contributing psychosocial factors [33]. There is evidence that surgical management using a spinal stimulator at a high frequency reduces symptom severity by half [57]. 

## 4. Conclusions

Although CECS and PAES are both known phenomena observed in collegiate athletes presenting with leg pain, their co-occurrence is uncommon; the successful diagnosis of one condition does not exclude the possibility of the existence of a second, unrelated diagnosis. A thorough history, physical, and evaluation must be conducted on patients presenting with common complaints in order to ensure an accurate diagnosis. We have provided a broad literature review of other diagnoses with the presenting symptom of bilateral leg pain, including external vascular compression, vessel wall disease, musculoskeletal overuse syndromes, and nervous system pathologies as they relate to the case presented. This case also highlights the importance of dynamic imaging modalities in assessing a wide range of diagnoses, including point-of-care ultrasound, dynamic MRA, and dynamic angiogram. 

## Figures and Tables

**Figure 1 diagnostics-14-01825-f001:**
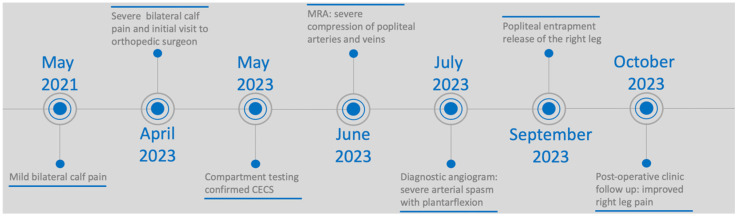
Timeline of patient’s symptoms, testing, and treatment.

**Figure 2 diagnostics-14-01825-f002:**
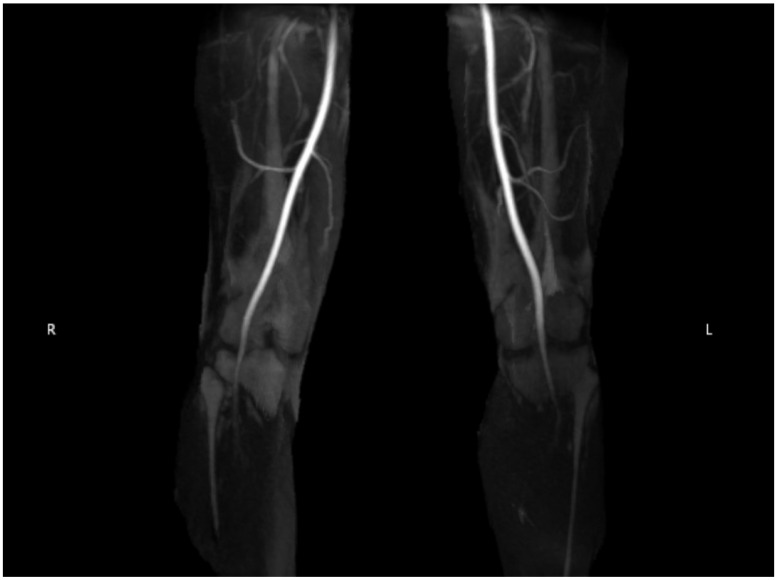
Magnetic resonance angiography of bilateral popliteal arteries.

**Figure 3 diagnostics-14-01825-f003:**
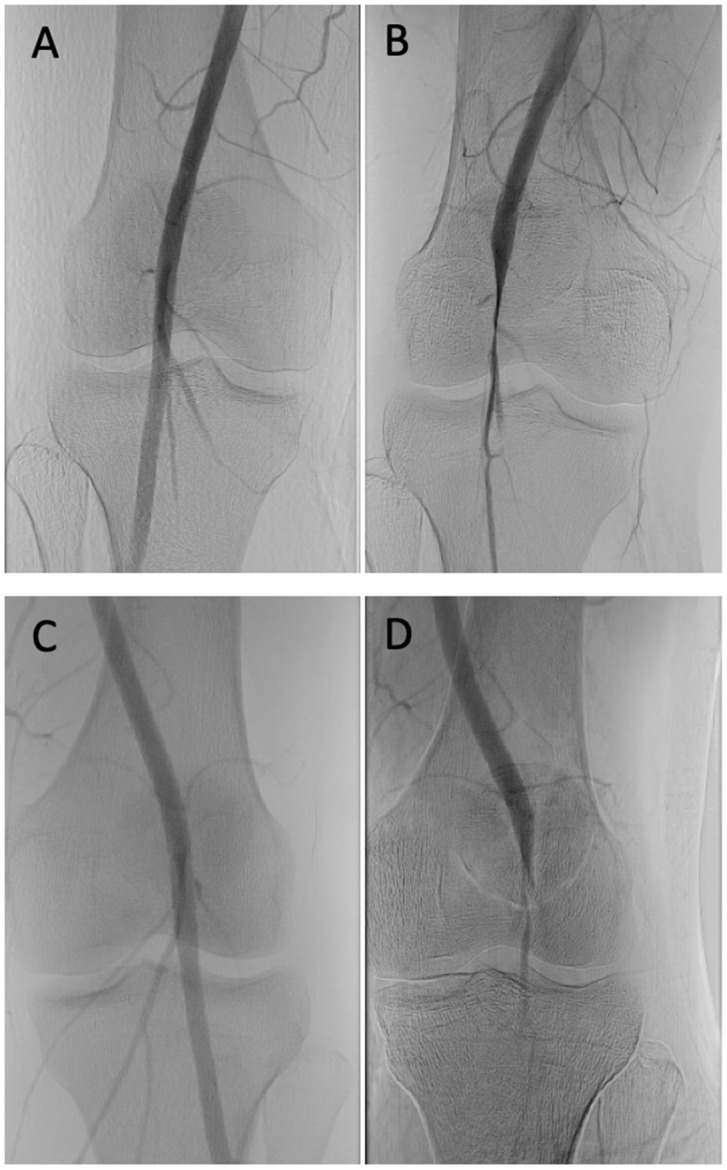
Diagnostic angiogram of bilateral popliteal arteries. (**A**) Right popliteal artery with normal flow. (**B**) Right popliteal artery compressed during plantarflexion. (**C**) Left popliteal artery with normal flow. (**D**) Left popliteal artery compressed during plantarflexion.

**Table 1 diagnostics-14-01825-t001:** Comparison of chronic exertional compartment syndrome (CECS) and popliteal artery entrapment syndrome (PAES) features.

	CECS	PAES
Epidemiology	Common (~30%)	Rare (~1%)
Mechanism	Increased intra-compartmental pressure within a fascial space.	Popliteal artery is compressed by the medial head of the gastrocnemius proximally and the fascial band of the soleus distally during activity.
Presentation	Exertional leg pain in any of the four compartments of the leg (anterior is most common). Other symptoms include swelling, cramping, and burning symptoms.	Exertional leg pain in the superficial posterior compartment of the leg. Other symptoms include cramping and tenseness in the posterior leg on palpation.
Laterality	Frequently bilateral.	Unilateral or bilateral.
Neurologic symptoms	Most commonly, peroneal nerve symptoms, although may have other neurologic compromise.	Rare paresthesias at the sole of the foot.
Physical Exam	After exercise, the affected compartment will be tender, tense, and painful to passive stretch.	Weaker pulses with the foot in dorsiflexion or plantarflexion and a drop in ankle brachial index of 30–50% with ankle dorsiflexion.
Diagnosis	Compartment testing > 30 mmHg one minute after ceasing pain-provoking exercises.	CTA, MRA, or diagnostic angiography for confirmation of suspected PAES.
Treatment	Conservative: physical therapy and massage, trialed for 3 to 6 months. Surgical: for failed conservative treatments, fasciotomy is recommended.	Surgical: open surgical decompression and release of entrapment of the medial head of the gastrocnemius or musculotendinous band.

## Data Availability

No new data were created or analyzed in this study. Data sharing is not applicable to this article.

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
