# Peer review of "Concurrent Chronic Exertional Compartment Syndrome and Popliteal Artery Entrapment Syndrome"

_diagnostics, 2024, doi:10.3390/diagnostics14161825_

Round 1

Reviewer 1 Report

Comments and Suggestions for Authors

Concurrent Chronic Exertional Compartment Syndrome and Popliteal Artery Entrapment Syndrome.

The manuscript under review is a case report of a unique instance of exertional leg pain caused by two concurrent entities in the same patient, namely, Chronic Exertional Compartment Syndrome (CECS) and Popliteal Artery Entrapment Syndrome (PAES). This case report presents a young female athlete who experienced bilateral exertional leg pain. After thorough evaluation, she was concurrently diagnosed with both CECS and type VI PAES. This dual diagnosis is unique.

About the report:

-          The introduction is concise and well-written. It provides sufficient background information to understand the significance of the conditions involved and the importance of accurate differential diagnosis.

-          Case Presentation: the case presentation is clear and well-structured. The patient's symptoms, and diagnostic process are well detailed. Table 1, which summarizes the differences between CECS and PAES, assists readers in understanding the two conditions.

-          The discussion section is well-organized and addresses the different etiologies of exertional leg pain. The citations are accurate and support the discussion. The authors provide a comprehensive overview of all potential causes of exertional leg pain. While the discussion is robust, I suggest including a section with prognostic aspects of patients with similar conditions: it would be beneficial to explore what is reported in the literature regarding the long-term outcomes for young athletes diagnosed with CECS or PAES, mainly, the possibility of returning to high-intensity sports post-surgery.

-          The conclusion is consistent with the overall report: it summarizes the key points and reinforces the significance of the case.

This manuscript presents a well-documented case report. Addressing the suggested areas for improvement would further enhance the manuscript's impact.

Author Response

Reviewer 1

The manuscript under review is a case report of a unique instance of exertional leg pain caused by two concurrent entities in the same patient, namely, Chronic Exertional Compartment Syndrome (CECS) and Popliteal Artery Entrapment Syndrome (PAES). This case report presents a young female athlete who experienced bilateral exertional leg pain. After thorough evaluation, she was concurrently diagnosed with both CECS and type VI PAES. This dual diagnosis is unique.

About the report:

-          The introduction is concise and well-written. It provides sufficient background information to understand the significance of the conditions involved and the importance of accurate differential diagnosis.

Author Response: We thank this reviewer for their supportive comments regarding the background information.

-          Case Presentation: the case presentation is clear and well-structured. The patient's symptoms, and diagnostic process are well detailed. Table 1, which summarizes the differences between CECS and PAES, assists readers in understanding the two conditions.

Author Response: We did put in a significant amount of effort in Table 1 in order to provide clarity for readers.

-          The discussion section is well-organized and addresses the different etiologies of exertional leg pain. The citations are accurate and support the discussion. The authors provide a comprehensive overview of all potential causes of exertional leg pain. While the discussion is robust, I suggest including a section with prognostic aspects of patients with similar conditions: it would be beneficial to explore what is reported in the literature regarding the long-term outcomes for young athletes diagnosed with CECS or PAES, mainly, the possibility of returning to high-intensity sports post-surgery.

Author Response: In our current manuscript, we have not mentioned long term outcomes for either CECS or PAES. We have now added an additional section to provide the information requested to our manuscript.

Manuscript Change:

Page 10-11: “The main treatment strategy for diseases within the external vascular compression category is to remove the external compressor. This can be done through either physical therapy or invasive surgical intervention. For CECS, initial treatment of gait retraining involves changing the patient’s pattern to a forefoot strike [8], [42]. Other conservative treatments involve rest and avoidance of the activity [43]. Given avoidance is not a favorable option for athletes and CECS can cause refractory symptoms, most patients undergo a fasciotomy to remove the external compression [9]. In a systematic review article outlining the outcomes of operative management, data from 24 studies including 1,596 patients were extracted and the outcomes were controversial [44]. The overall complications consisted of neuritis, infection, and hematoma at a rate of 13%. The rate of symptom improvement after compartment release for CECS was 66% and 84% of patients were satisfied with their surgical outcomes in follow up. Explanations for unsuccessful cases included no consensus on the definition of “success” and a subset of patients with deep posterior compartment syndrome may derive less benefit from surgical release [45].

Conversely, initial management of symptomatic PAES is surgical exploration and release of the popliteal artery through sectioning of fibrous bands [16], [46]. Left untreated, PAES may result in complications such as popliteal artery stenosis, thrombosis, or limb-threatening distal arterial thromboembolism. For PAES, surgical outcomes are generally positive: one recent retrospective study reported no complications and a success rate of 100% [47]. There is a higher rate of symptom improvement in PAES symptoms compared to CECS at 97% [48], [49]. Complications such as postoperative hematoma, poor healing, and infections occurred in up to 16% of patients [48]. Long-term follow-up studies indicate that the primary patency rate is 98%. A significant portion of patients are able to resume sports, with about 33% returning fully, 50% partially, and 17% not resuming sports at the time of follow up [50].

Vessel wall diseases are normally treated with aggressive conservative strategies. PAD is treated with a rigorous walking therapy and risk factor modification [51]. If the disease becomes severe and the patient develops rest pain or an ulcer, an endovascular first approach is generally used [52]. The prognosis of PAD overall varies widely depending on compliance, comorbidities, and disease severity. Severe PAD requiring amputation confers increased risk of mortality, with around 50% of patients dying within one year of their amputation [53]. While arterial endofibrosis is also a vessel wall disease normally treated conservatively with position modification, this disease occurs in healthy cyclists [54]. The pathology is related to repetitive use and most patients have a good prognosis [55].

            The most common musculoskeletal overuse disease is MTSS, where a daily regimen of calf stretching and eccentric calf exercises is highly recommended and efficacious [56]. Other recommended treatments include shock absorbing soles of shoes, insoles to prevent excessive foot pronation, and proprioceptive balance [57]. For patients with refractory disease, surgical treatment performed by removing a strip of periosteum produced an “excellent” result in only 35% of patients [58].

The treatment of leg pain originating from nervous system dysfunction varies widely depending on pathology. While there is limited high level evidence of a variety conservative treatments, expert consensus agrees that a limited course of structured exercise is warranted for patients with lumbar disc herniation with radiculopathy [32]. There is a moderate level of evidence that supports discectomy for lumbar disc herniation with radiculopathy in patients whose straight leg test is positive [32]. Pain as measured by the visual analogue scale 6 months post-operatively improved in over 70% of patients [59][60]. Treatments for nerve entrapments are mainly conservative: discontinuation of aggravating activities, use of local physical modalities such as ice, heat, or transcutaneous electrical nerve stimulation; desensitization techniques, correction of training errors that may be contributing to the nerve entrapment, braces, and nonsteroidal anti-inflammatory drugs may all be used [33]. The surgical approach for patients with refractory symptoms depends on the type of nerve entrapment. For peroneal nerve entrapment requiring surgical decompression, athletes usually return to normal activities of daily living within 3 weeks post-operatively [61]. Complex Regional Pain Syndrome is another nervous system disorder that is primary treated by targeting both exercise and contributing psychosocial factors [35]. There is evidence that surgical management using a spinal stimulator at high-frequency reduces symptom severity by half [62].”

-          The conclusion is consistent with the overall report: it summarizes the key points and reinforces the significance of the case.

Author Response: We thank this reviewer for acknowledging the significance of this case.

Reviewer 2 Report

Comments and Suggestions for Authors

Dear colleagues!

Issues of pain and blood supply today are acute at the border of science and practical healthcare, making your research relevant.

I would like to focus on high-quality presentation of data, competent design and methodical presentation of information.

However, I would like to share my comments:

1. In the introduction, it will be useful to expand information about the epidemiology of the disease, the quality of medical care and diagnosis.

2. It is advisable to add information to the materials and methods about objective methods for assessing pain, as well as about the procedure for the radiation method of research (dosage, equipment)

3. I don’t quite understand the necessity of the section “5. Patient Perspective”. Please write an explanation

Author Response

Reviewer 2

Issues of pain and blood supply today are acute at the border of science and practical healthcare, making your research relevant.

I would like to focus on high-quality presentation of data, competent design and methodical presentation of information.

However, I would like to share my comments:

  1. In the introduction, it will be useful to expand information about the epidemiology of the disease, the quality of medical care and diagnosis.

Author Response: We have now included a brief overview of the epidemiology and diagnoses strategies in the introduction. Detailed information on the quality of medical care and diagnosis strategies can be found under Discussion section 3 in the subsections corresponding to each category of disease related to lower extremity leg pain.  

Manuscript Change:

Page 1 Lines 33-41: “With regards to incidence, CECS has been described as occurring in up to 0.49 cases per 1,000 persons-years [4]. PAES is even more rare at an incidence of 0.16 to 3.5 cases per 100,000 person years [5]. Diagnostic approaches of these diseases include physical exam ma-neuvers and various imaging modalities. In this specific case, dynamic imaging mo-dalities were the most elucidating in the assessment of external vascular compression, vessel wall disease, and musculoskeletal overuse syndromes. While static computed tomography angiography (CTA) was the prior screening and diagnostic tool of choice to assess for compression, ultrasound imaging, dynamic angiography, and dynamic MRA have proven their utility.”

  1. It is advisable to add information to the materials and methods about objective methods for assessing pain, as well as about the procedure for the radiation method of research (dosage, equipment)

Author Response: Although there is no materials and methods section in this case-based literature review, we have added the standard radiation exposure for CTA in the text. We do not focus on objective methods for assessing pain and instead focus our attention on a detailed description of each pathology in the discussion section,

Manuscript Change:

Page 10 Lines 326-327: “Radiation exposure is similar to a static CTA, exposing the patient on average to less than 4 miligray [41].”

  1. I don’t quite understand the necessity of the section “5. Patient Perspective”. Please write an explanation

Author Response: On original request, we included information on the patient perspective. Given this reviewer does not feel it is necessary, we have removed it from the manuscript.

Manuscript Change: Section 5 was removed from the manuscript.